# A Natural Experiment Comparing the Effectiveness of the “Healthy Eagles” Child Weight Management Intervention in School Versus Community Settings

**DOI:** 10.3390/nu13113912

**Published:** 2021-10-31

**Authors:** Melissa Little, Shirley Serber-Souza, Maryam Kebbe, Paul N. Aveyard, Susan A. Jebb

**Affiliations:** 1Nuffield Department of Primary Care Health Sciences, University of Oxford, Radcliffe Primary Care Building, Radcliffe Observatory Quarter, Woodstock Road, Oxford OX2 6GG, UK; maryam.kebbe@pbrc.edu (M.K.); paul.aveyard@phc.ox.ac.uk (P.N.A.); susan.jebb@phc.ox.ac.uk (S.A.J.); 2Nutrition and Dietetics, Foodtalk CIC, London SE16 6SG, UK; shirley@foodtalk.org.uk; 3Reproductive Endocrinology and Women’s Health Laboratory, Baton Rouge, LA 70802, USA

**Keywords:** childhood obesity, weight management, children, behavioural weight management interventions

## Abstract

Behavioural weight management interventions are recommended for the treatment of obesity in children. However, the evidence for these is limited and often generated under trial conditions with White, middle-class populations. Healthy Eagles is a behavioural weight management intervention designed to treat excess weight in children. It ran in the London Borough of Croydon from 2017 to 2020 and was delivered in both school and community settings, providing a natural experiment to compare outcomes. A total of 1560 participants started the Healthy Eagles programme; 347 were in the community setting and 703 in the school setting. Data were analysed for those who completed 70% of the programme. In the school setting, there was a small but significant reduction in BMI z-score (M = −0.04, 95% CI = −0.08, −0.01) for participants above a healthy weight, especially in those with severe obesity (M = −0.09, 95% CI = −0.15, −0.03); there was no significant change in any subgroup in the community setting. Linear regression analysis showed the school setting was associated with a 0.26 (95% CI = 0.13, 0.49) greater reduction in BMI z-score than the community setting after adjusting for ethnicity, deprivation, age and gender. Across both programmes, the effect was somewhat greater in participants from a Black (African/Caribbean/Other) ethnic background (M = −0.06, 95% CI = −0.09, −0.02) and from the two most deprived quintiles (M = −0.06, 95% CI = −0.11, −0.01). Data were limited, but minimal changes were measured in nutrition and physical activity behaviours regardless of setting. This evaluation provides indirect evidence of a small but significant benefit to running weight management interventions in a school versus community setting.

## 1. Introduction

In England, over one in five reception age children (aged 4–5) and over one in three year six children (aged 10–11) are above a healthy weight [1]. Excess weight in childhood leads to adverse physical and psychosocial health consequences, often persists into adulthood, and is associated with morbidity and early mortality [2,3]. Exposure to excess weight in childhood is a risk factor for a number of non-communicable chronic diseases in adulthood and represents an additional economic strain on health services [4]. Effective interventions to prevent and treat obesity in childhood are urgently needed.

The National Institute for Health and Care Excellence (NICE) recommend family-based lifestyle and behavioural weight management interventions to treat excess weight in children [5]. These are 10–12-week family-based interventions run in a community setting and incorporating nutrition, physical activity and behaviour change counselling. NICE suggests that interventions should target only children above a healthy weight and that participants should attend at least once per week for a minimum of one hour [5]. However, the evidence to support these interventions is limited and, in most cases, is generated under trial conditions, often with a preponderance of White, middle-class populations [6]. As obesity is a condition that disproportionately affects minority ethnic and deprived populations [1], the evidence does not currently reflect those most at risk. Additionally, evidence generated under trial conditions may not take into account the challenges faced when implementing an intervention in a real world setting. Natural experiments, which evaluate interventions occurring within existing public health practice, often provide important evidence of the true effectiveness of interventions that can inform health policies and guidance. These pragmatic study designs can reveal the realities of implementation and delivery which may hamper intervention effectiveness [7,8]

Healthy Eagles is a childhood behavioural weight management intervention that aims to treat obesity in children aged 4–16 by increasing positive health behaviours, ultimately leading to maintenance or reduction of body mass index (BMI) z-score in children. Healthy Eagles was delivered in the London Borough of Croydon from December 2017 to June 2020 and ran in two different settings; community settings such as leisure centres or community halls, and secondary schools. Systematic reviews assessing the effectiveness of obesity treatment interventions in childhood exist for both school [9,10,11] and community [12,13] settings. However, the evidence for both is mixed with neither setting showing strong evidence of long-term BMI z-score reduction. Most recently a systematic review, meta-analysis and meta-regression [14] suggested that school-based interventions have the potential to treat obesity but the results of the review were inconclusive due to a lack of high-quality evidence. Kelleher et al. [15] showed similar potential in community interventions but cited barriers to success such as low uptake and high attrition rates. 

By running the Healthy Eagles programme simultaneously across both settings it created the opportunity to directly compare programme outcomes.

Additionally, although the programme content was the same in both settings, the community venues included only children above a healthy weight and the secondary schools included all children regardless of weight. As the school-based programme was run with participants of all BMI z-scores, it became possible to explore the programme effect in participants with a healthy weight as well as those with excess weight. 

The primary aim of this natural experiment was to assess the change in BMI z-score, and associated health behaviours, of children who attended the Healthy Eagles child weight management programme and to compare these changes in the school versus the community setting. The secondary aim was to examine what, if any, outcome differences were observed in relation to gender, ethnicity, age or baseline weight status.

## 2. Materials and Methods

### 2.1. Study Design

This was an evaluation of a natural experiment of a child weight management intervention delivered in two different settings; secondary schools and community venues. Data were collected from participants attending the Healthy Eagles programme between December 2017 and June 2020. As this was a natural experiment data had already been collected by the programme provider and participants had consented to take part in the intervention and have their data collected for the purposes of programme evaluation. No additional consent for this academic evaluation was required [16].

### 2.2. Intervention

Healthy Eagles was designed and developed by Foodtalk, a dietitian-led Community Interest Company. The programme was conceived based on NICE guidance and on emerging and promising research in treating excess weight in children [10,12,15]. The programme’s underlying assumptions are described in Table 1. 

The Healthy Eagles programme is underpinned by the Social Cognitive Theory of behaviour change [17]. The key underlying concepts were self-efficacy, reinforcement, behavioural capability, and observational learning. To improve self-efficacy and behavioural capabilities, participants were provided with the skills and knowledge to achieve their goals and positive reinforcement was used to celebrate achievements. Goals and rewards were self-led, allowing participants to define both outcomes and expectations of the programme. Observational learning was practised by allowing participants to observe both the coaches and their peers and engage in health behaviour modelling of nutrition and physical activity behaviours.

The intervention itself consisted of ten weekly sessions ranging from 1–1.5 h each. Topics, session layout and key methods are described in Table 1. Core components based on behaviour change were repeated each week and included goal setting and reflection, food tasting and mindfulness. Although core components and session topics remained the same across the intervention, methods of delivery were adjusted based on age, such as drawing goals for younger children and writing them for older children. Participants were also given a guided journal and asked to fill this in weekly to support their learning at home. Parental involvement was encouraged for participants under the age of eleven (*n* = 26) however for the majority of participants (*n* = 509) that were aged eleven or older, there was no parental involvement. 

The Palace for Life Foundation (PFL), the charitable arm of Crystal Palace Football Club (FC), a Premier League Football team, led the delivery. Crystal Palace FC is a stalwart of the Croydon community and has an immediate name and brand recognition throughout the borough and an extensive social media following, especially amongst the younger Croydon generation. By combining Foodtalk’s expertise in child weight management with the ground level, borough-wide knowledge of PFL, an intervention was created based on evidence and, by using local signposting and localised resources, made bespoke to the needs of the Croydon community. 

Healthy Eagles, in both the school and community settings, was delivered by health coaches, most with a physical activity or nutrition background, who were hired and managed by PFL. Coaches received intensive two-day training delivered by Foodtalk and a comprehensive programme delivery manual outlining key topics, background reading, and a step by step guide to each session. For quality assurance, each coach was supervised during delivery once per term, and a WhatsApp group was set up to provide continuous ongoing support to coaches. As recommended by the Medical Research Council [18], an extensive process review was undertaken during the second year of the programme to ensure programme fidelity and determine any key barriers and facilitators to fidelity maintenance. This was done through structured observations, focus groups with coaches, self-reported participant questionnaires and an audit of participant attrition. The process evaluation found that whilst most programmes were delivered with contextual or content modifications, this did not significantly impact participant learning outcomes. Additionally, by identifying key barriers to fidelity, the team could address these and make improvements.

Healthy Eagles was initially designed to be delivered in community settings. However, later this was adjusted to include delivery in schools alongside the community settings. There was no difference in content, method of delivery or delivery staff between the school and community programmes. The only difference was that the school programme included all participants regardless of weight and the community programme only included participants above a healthy weight. Additionally, the community programme offered a “rolling, drop-in” design, meaning that participants could start immediately after referral instead of waiting for an appropriate programme to begin. This offered flexibility for participants unable to attend every week and had the added benefit of allowing participants just starting their journey to learn from those further along in the programme creating a peer-to-peer mentoring model. Additionally, as the programme continued even after participants “graduated”, they could drop back in at any point if they felt they needed extra support. When delivered in schools, the programme reverted to concurrent weekly sessions. As attendance was mandatory and during school hours, it was unnecessary to provide a flexible model.

### 2.3. Participants

Children were eligible for the programme if they were aged 4–16 years and either living in the London Borough of Croydon or attending a school in the borough. The community programme’s eligibility criteria were BMI greater than 91st Centile (BMI z-score > 1.34), considered above a healthy weight in the World Health Organisation Growth Reference Charts 2007. The school programme had no BMI criteria and thus included participants with a range of BMI z-scores. There were no exclusion criteria. 

Participants were recruited through the National Child Measurement Programme (NCMP), self-referred or referred by GPs, health care professionals or their school. This was accompanied by extensive Healthy Weight Awareness training throughout the borough with 550 school, health and voluntary staff receiving training on raising the issue of weight with families and how to refer into the programme. Working with the broader community of health and care professionals aimed to increase awareness of the complexity of obesity, challenge the stigma surrounding obesity and improve the way the issue of weight was raised with families. 

The in-school programmes targeted secondary schools in areas with high levels of deprivation and obesity based on the NCMP results and deprivations maps. Letters were sent to parents informing them of the programme and asking them to contact the Healthy Eagles team if they had questions. In most schools, the Healthy Eagles programme was delivered as part of a Personal Social and Health Education (PSHE) lesson or a Physical Education (PE) lesson and often, it was delivered to students who refused to engage in standard PE provision. For all school-based programmes, schools decided which classes and year groups would participate. 

### 2.4. Outcomes Measures

The primary outcome was the change in BMI z-score from baseline to post-intervention. Weight and height measurements were taken at the beginning and end of the intervention by programme delivery staff after comprehensive training and following measurement guidelines for the NCMP [19]. Measurements were taken without shoes in light clothing and recorded to the nearest 0.1 cm. Raw data on height, weight, date of birth (DOB) and date of measurements (at baseline and post-intervention) were used to calculate BMI of children and converted to BMI z-scores based on the 1990 UK Growth Reference curves [20] using LMS Growth Software (Pan and Cole). BMI z-scores allow for a single measure relevant across gender and age and over time [5]. 

The secondary outcomes were changes in six health behaviours chosen to reflect behavioural goals within the programme and relevant to weight control [21]. These were consumption of vegetables, consumption of sweet and/or savoury snacks, physical activity levels, sedentary behaviour, eating together as a family, and eating in front of the television. Health behaviours were assessed through self-reported non-validated questionnaires given on the first and last day of the intervention; participants under 11 years had parental support for completion Nutrition and eating behaviours were assessed by asking “On most days when do you do/eat the following?” with meal and snack times listed below for participants to tick. Physical activity and sedentary behaviour were assessed by asking participants to tick approximately how much activity or sedentary behaviour they have on each weekday and weekend with answers divided into six groups ranging from “less than 1 h” to “over 8 h”.

Demographic data, including participant age, gender, ethnicity and postcode, were collected at the first point of contact. Postcode was used to evaluate deprivation decile based on the English Indices for Multiple Deprivation 2019 [22]. 

### 2.5. Analysis

#### 2.5.1. Data Cleaning 

Data were checked for plausibility, and highly improbable measures, such as significant reductions in height, were removed from the analysis. Data were only analysed for participants who attended at least seven of the ten programme sessions and had complete pre and post-intervention data. Data were not analysed on an intent-to-treat basis because adherence differed considerably between the school and community programme, with the school taking an opt-out rather than opt-in approach [23].

#### 2.5.2. Primary Outcome 

The analysis used data from all eligible participants in both school and community settings with complete data on BMI. A paired t-test was done on BMI z-score using pre and post-intervention measures. 

Subsequently, participants were divided into groups based on programme setting (school and community) and baseline BMI z-score. The following z-score cut-offs were used to define weight categories based on the British 1990 cut-offs for BMI [20]; healthy weight (z-score −1.34 to 1.34), overweight (z-score 1.35 to 2.05), obesity (z-score 2.06 to 2.65), severe obesity (z-score > 2.65).

Multiple linear regression was used to compare outcomes between the school-based and community-based intervention with the community-based intervention as the reference. This analysis was restricted to children above a healthy weight as the community programme did not include healthy weight participants. The analysis was adjusted for differences in socioeconomic status, age, gender, and ethnic group. 

Among participants above a healthy weight, we conducted subgroup analysis for groups defined by gender, ethnicity, age and deprivation using a paired *t*-test calculation. Ethnicity was divided into four sub-categories: White (British/Irish/Other), Black (African/Caribbean/Other), Asian, and any other ethnic background. Age was divided into older and equal to, or younger than, 11 years. Gender was divided into girls and boys, and deprivation was sub-categorised into the two most deprived and two least deprived quintiles.

#### 2.5.3. Secondary Outcomes

This analysis was based on changes measured in questionnaires completed at baseline and end of the programme. For physical activity and sedentary behaviour, hours per day were rounded to the highest whole hour (i.e., 1–2 h was rounded to 2 h) and calculated to give a total number of weekly hours. Dietary intake and mealtime behaviours were converted to numeric scores where 0 = never and 5 = at every meal and snack. Scores were analysed using a paired t-test calculation with baseline and post-intervention data. 

All calculations were done using SPSS for IBM Version 27 and we took an estimation approach rather than confirmatory hypothesis testing approach, with confidence intervals set at 95%.

## 3. Results

A total of 1087 participants completed the Healthy Eagles programme between December 2017 and June 2020, meaning they attended at least seven of the ten sessions. Completion of the school-based programme was 100% compared to 32.7% in the community-based programme. A comparison of baseline characteristics showed no significant differences in gender, age, deprivation or ethnicity for completers versus non-completers in the community-based programme.

Of the 1087 participants who completed the programme, 535 participants (49%) were deemed eligible for the analysis, meaning they had all the data needed to calculate both pre and post-intervention BMI z-score, including gender, date of birth, height (pre/post), weight (pre/post) and date of pre and post measurements (Figure 1). The school programme had the highest percentage of participants with complete data at 74.3% for participants above a healthy weight and 43.1% for healthy weight participants. The community programme had 24.8% of participants with complete data.

Table 2 shows the participant demographic characteristics at baseline, first divided by setting and then further divided by baseline weight status. Two children who were underweight were excluded from the study as the sample size for this subgroup was too small to give meaningful results. 

### 3.1. Intervention Outcomes 

Overall, there was no evidence of a change in BMI z-score for participants above a healthy weight from baseline (M = 2.36) to post-intervention (M = 2.33) (mean difference = −0.03, 95% CI = 0.00, −0.06). In subgroup analyses of all participants above a healthy weight, there was evidence for a benefit only in Black (African/Caribbean/other) ethnicity, children older than 11, and those from the most deprived two quintiles (Table 3). 

### 3.2. Effects of Setting on BMI z-Score in Children above a Healthy Weight

In the school programme, there was a reduction in BMI z-score for children above a healthy weight (M = −0.04, 95% CI = −0.08, −0.01) with the greatest reduction in children identified with severe obesity (M = −0.09, 95% CI= −0.15, −0.03) (Table 4).

In the community programmes, the sample size was much smaller and there was no evidence of a change in BMI z-score for the group as a whole or in any subgroups. Indeed, the trend was towards an increase (Table 4). Linear regression analysis showed that the school setting was associated with a 0.24 (95% CI: 0.10, 0.44) greater reduction in BMI z-score (unadjusted), and a 0.26 (95% CI: 0.13, 0.49) greater reduction in BMI z-score when adjusted for ethnicity, deprivation, age and gender.

### 3.3. Secondary Outcomes

In general, there was no evidence of changes in health behaviours regardless of baseline BMI z-score or setting. Table 5 shows the overall change in health behaviours from pre- to post-intervention based on self-reported questionnaires completed by the participants. Not all participants included in the analysis had complete data for each question therefore N refers to the number of participants with complete data that were included in the analysis of each question. There was limited evidence that participants in the community programme reduced the frequency in which they ate while watching television, however, there was also limited evidence to show that they also decreased their self-reported physical activity.

## 4. Discussion

### 4.1. Summary

This natural experiment aimed to assess the change in BMI z-score of children who attended the Healthy Eagles child weight management programme and to compare these changes in the school versus the community setting. Overall, the intervention showed no evidence of a difference in BMI z-score for children above a healthy weight from pre to post-intervention. Results from a subgroup analysis showed evidence of a reduction in BMI z-score for children of Black (African/Caribbean/other) ethnicity, those over the age of 11 and those from the most deprived two quintiles; all other subgroups showed no evidence of a change. Analysis of the two settings showed that children above a healthy weight in the school setting significantly decreased BMI z-score with BMI z-score in the community setting trending upwards. There was no BMI z-score change in children identified as a healthy weight. All children in the school cohort completed the intervention, and three quarters had data included in the primary analysis; only 33% did so in the community cohort, with 25% included in the analysis. There was little evidence of change in health behaviours.

A Cochrane review on lifestyle interventions for treating obesity in children shows that this type of behavioural weight programme may achieve small effects in the short term [24], However, long-term success is minimal, and the quality of evidence is limited. Whilst the school-based intervention in our study showed a small but significant reduction in BMI z-score in children identified as having severe obesity, the study cannot assess whether this was as a direct result of the intervention or whether these changes would be sustained in the long term. 

### 4.2. Strengths/Limitations

This is the first evaluation to compare a weight management intervention across school and community settings. Although the comparison is not randomised or direct, the programme components and delivery team were the same, and the population was similar. One of the strengths of this investigation was the diversity in the participants. Although it is well established that childhood obesity adversely affects children from minority ethnic and/or deprived backgrounds, most studies of similar interventions still have a majority of participants from White middle-class backgrounds [25,26]. Eighty-three per cent of the participants in this investigation came from a minority ethnic background, and 63% came from the two most deprived quintiles. 

Another strength is that this research evaluated an extant programme, typical of those employed in public health practice in Britain. Currently, there is very little evidence of the effectiveness of interventions to treat obesity in routinely commissioned programmes, and available research shows that impact can vary greatly when compared to trial conditions [27,28]. This study shows the real-world effectiveness of a behavioural weight management intervention designed to meet NICE good practise guidance [8]. 

This study has limitations, many of which are inherent to natural experiments, namely, the lack of a control group for comparison. The results showed evidence of a reduction in BMI z-score for participants with severe obesity on the school programme; however, this was not accompanied by changes in behaviour expected to support weight loss. Incomplete or unreliable questionnaire data could explain these findings, or it is possible that BMI z-score change reflects regression to the mean, where participants with the highest BMI would typically expect to see a reduction at follow-up. However, randomised controlled trials conducted in similar populations often show a modest increase in BMI z-score in the control group [29,30] and results from the NCMP over a similar time period shows that, without intervention, the population average weight of children is increasing, especially in the most deprived communities [1]. Together, these support the hypothesis that the changes may have occurred because of the programme.

Observational analyses are also limited by residual confounding [8]. When comparing the two settings, we endeavoured to adjust for age, gender, deprivation, and ethnicity, but it is impossible to eliminate all other potential confounding variables. Additionally, because we used data from completers only rather than an intention to treat basis, bias due to loss to follow-up could affect findings. However, follow up in the school group was high and the greater loss to follow up in the community cohort would be expected to bias the findings towards children experiencing more successful outcomes from treatment; yet the outcomes still favour the school setting [23]. 

The data were collected in a setting where the priority was delivering the programme, not data collection and, as such, the data were often incomplete or missing. The community cohort had a much lower retention rate than the school cohort, 32.7% vs. 100%, respectively, reflecting the “opt-out” approach of the school intervention. The community programme also had a much lower percentage of usable data than the school programme, 24.8% vs. 74.3%, respectively. The most likely explanation is that the “rolling” nature of the community programme, in comparison to the “fixed” nature of the school programme, meant measurements or dates were sometimes missed by staff and BMI z-score could not be calculated. A final limitation is that the changes were examined over 10 weeks only, and it was not possible to assess the longer-term outcomes in this programme evaluation. 

### 4.3. Interpretation of Findings 

There was evidence that the BMI z-score decreased to a greater extent in the school-based programme than in the community-based programme. These findings differ from previous research which describes community settings as an integral component for success on child weight management interventions and school-based programmes having little to no effect [31,32]. However, many school-based studies are conducted in younger, primary school cohorts who may have less agency over their health behaviours than secondary school students and therefore are more likely to require parental involvement for success [33]. 

Although both intervention settings provided similar content, the community-based programme was tailored to only those above a healthy weight and required participants to “opt-in” to attendance, indicating a baseline level of motivation to change. This created an expectation that the community programme would prove more effective, however, we found no evidence of this. A review by Cui et al. [34] found some evidence that school-based settings may be more effective for retaining ethnic minority communities and people living in areas of high deprivation which is reflected in our study population. There is evidence that suggests it is the short but consistent nature of school-based programmes that contributes to their effectiveness with these groups [35] while other evidence indicates that it is the population level approach to delivery that leads to improved outcomes [36]. As the two settings differed in their inclusion criteria, it is difficult to say whether the increased effectiveness of the school programme is a function of the setting or population level approach. 

It is important to note the changes, or lack thereof, in health behaviours. Our analysis showed a significant decrease in physical activity in both community-based participants and participants of a healthy weight. It also showed no change in health behaviours for overweight, school-based participants despite significant BMI z-score reductions in this group. These inconsistencies may be a reflection of the pragmatic approach. We used very ‘light-touch’ questionnaires to capture the data in these ‘real-world’ settings which may not be sufficiently precise to capture small changes in behaviour. Additionally, the nutrition questions measured markers of diet quality rather than energy intake and therefore cannot be directly linked to weight change. Still, this finding is surprising and reinforces concerns about the effectiveness of these programmes.

The school-based Healthy Eagles programmes are easier to run and, because they do not require paying for a community venue, often cost less per participant. Hence, community programmes need to be more clinically effective to be more cost-effective. We found no evidence of this here. School-based programmes also recruit a much broader sample of participants, including those with the highest level of need who may not be able to attend a programme in the community. They offer support in attaining a healthier lifestyle for participants across a range of BMI z-scores, and we found no evidence of inappropriate changes in participants within a healthy BMI z-score range. If universal approaches can be as successful at reducing unhealthy weight as targeted interventions, they may provide an acceptable, and possibly preferable, alternative for weight management with less potential for stigma among children who have overweight and are referred to a community service [37].

## 5. Conclusions

This evaluation suggests a small but significant benefit to running weight management interventions in school versus in community settings. Improvements in BMI in the school-based programme were modest but proportionate to baseline BMI and with significantly greater effectiveness in children from Black (African/Caribbean/other) backgrounds and those from more deprived areas, suggesting some potential to target those most at risk of excess weight. 

## Figures and Tables

**Figure 1 nutrients-13-03912-f001:**
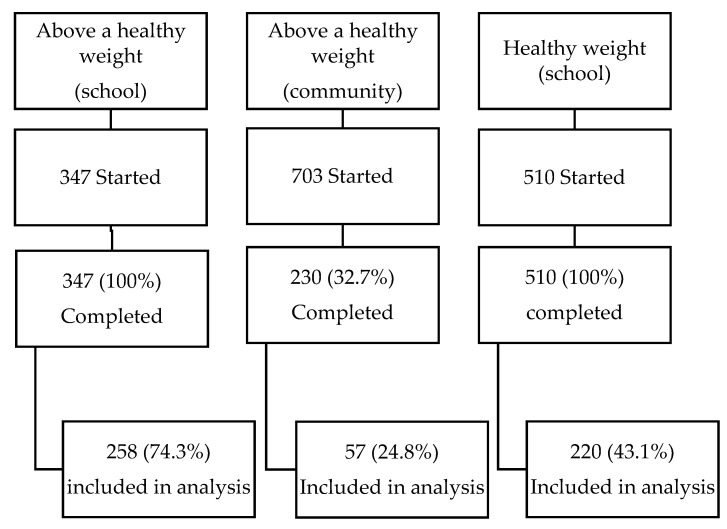
Participant flow chart.

**Table 1 nutrients-13-03912-t001:** Healthy Eagles programme session outline.

Session Topics	Session Format and Key Methods	
Introduction/ Measurement	Welcome	
Healthy Habits and Physical Activity	Review progress on goals	Participants were encouraged to write (older children), draw (younger children) or verbally share (both) their progress.
Food Groups and Portion Sizes	Introduce today’s topic	
Understanding Food Labels	Session Activities	Session activities were done through active or interactive games such as food groups relays or label reading challenges.
Sugar	Healthy Snack	This was prepared and eaten by participants with support from the coaches.
Takeaways and Fast Food	Goal Setting	Participants were encouraged to write (older children), draw (younger children) or verbally share (both) their goals.
Supermarket Tour	Journal Assignment	Guided journals were provided for the participants with activities (younger children) or guided writing tasks (older children).
Dieting: Fool or Wise?	Mindfulness and Reflection	Participants were encouraged to quietly reflect on where they could reflect on their journey thus far and where they see themselves in the future.
The Importance of You		
Maintenance and Signposting		
Underlying Assumptions 1. Programmes designed and delivered locally will have increased participant reach. 2. Providing both knowledge and behaviour change techniques will lead to improved participant outcomes. 3. Participants are more likely to return if the programme is enjoyable and interactive. 4. By encouraging participants to set personal goals and reflect on them they are more likely to take ownership in achieving those goals.		

**Table 2 nutrients-13-03912-t002:** Participant characteristics at baseline.

	Total	School				Community				
		All	Healthy Weight	Over Weight	Obesity	Severe Obesity	All	Over Weight	Obesity	Severe Obesity
Baseline BMI z-score Mean (SD)		1.3 (1.41)	0.1 (1.05)	1.71 (0.21)	2.32 (0.17)	3.32 (0.81)	2.44 (0.68)	1.68 (0.18)	2.31 (0.18)	3.11 (0.46)
n	535	478	220	116	65	77	57	17	17	23
Gender *n* (%)										
Girl	323 (62)	322 (67)	160 (73)	78 (67)	41 (63)	43 (56)	31 (54)	13 (76)	7 (41)	11 (48)
Boy	201 (38)	156 (33)	60 (27)	38 (33)	24 (27)	34 (44)	26 (46)	4 (24)	10 (59)	12 (52)
Ethnicity *n* (%)										
Black (African/ Carribean/ Other)	231 (48)	225 (47)	119 (54)	48 (41)	33(51)	25 (32)	30 (53)	10 (59)	10 (59)	10 (43)
White (British/ Irish/ Other)	70 (15)	64 (13)	26 (12)	16(14)	9 (14)	13 (17)	8 (14)	2 (12)	2 (12)	4 (17)
Asian	80 (17)	91 (19)	40 (18)	28 (24)	10 (15)	13 (17)	9 (16)	3 (18)	3 (18)	3 (13)
Mixed	22 (5)	46 (10)	9 (4)	18 (16)	7 (11)	12 (16)	9 (16)	2 (12)	2 (12)	5 (22)
Unknown	78 (16)	52 (11)	26 (12)	6 (5)	6 (9)	14 (18)	1 (2)	0	0	1 (4)
Deprivation Decile * Mean(SD); unknown	3.72 (1.08)	3.66 (1.11)	3.62 (1.08)	3.51 (1.26)	3.68 (1.09)	3.94 (0.99)	4.05 (0.77)	3.88 (0.72)	4.13 (0.96)	4.17 (0.65)
Age Mean (SD)	11.7 (2.07)	12.04 (1.83)	12.42 (1.38)	12.16 (1.97)	11.91 (1.50)	11.04 (2.28)	10.6 (2.60)	11.18 (3.21)	10.71 (1.79)	10.46 (2.54)

* 1 is most deprived, 10 most affluent.

**Table 3 nutrients-13-03912-t003:** Subgroup analysis of participants above a healthy weight by ethnicity, age, gender and level of deprivation.

	N	Mean Change in BMI z-Score Pre/Post Intervention (95% CI)
All participants above a healthy weight	315	
White (British/Irish/Other)	46	0.03 (−0.10, 0.15)
Black (African/Caribbean/other) *	136	−0.06 (−0.09, −0.02)
Asian	49	−0.00 (−0.08, 0.08)
Any other ethnic background	45	−0.06 (−0.17, 0.05)
<11	89	0.01 (−0.08, 0.09)
>11 *	124	−0.10 (−0.14, −0.06)
Girl	193	−0.02 (−0.06, 0.01)
Boy	121	−0.05 (−0.12, 0.01)
Most deprived 2 quintiles *	121	−0.06 (−0.11, −0.01)
Least deprived 2 quintiles	33	−0.10 (−0.25, 0.04)

* indicates *p*-value of <0.05.

**Table 4 nutrients-13-03912-t004:** Change in BMI z-score pre and post-intervention by setting and baseline BMI z-score.

	N	Mean change in BMI z-Score Pre/Post Intervention (95% CI)
All participants above a healthy weight	315	−0.03 (−0.64 to 0.01)
All healthy weight participants	220	0.03 (−0.03 to 0.09)
School Setting		
All participants above a healthy weight *	258	−0.04 (−0.08 to −0.01)
Only participants who are overweight	115	−0.01 (−0.07 to 0.05)
Only participants with obesity	66	−0.05 (−0.13 to 0.03)
Only participants with severe obesity *	77	−0.09 (−0.15 to −0.03)
Community Setting		
All participants above a healthy weight	57	0.03 (−0.06 to 0.12)
Only participants who are overweight	17	−0.09 (−0.26 to 0.09)
Only participants with obesity	17	0.14 (−0.11 to 0.39)
Only participants with severe obesity	23	0.03 (−0.03 to 0.10)

* indicates *p*-value of <0.05.

**Table 5 nutrients-13-03912-t005:** Mean change in health behaviours from baseline to post-intervention, by baseline BMI z-score and setting.

	N	Mean Change Pre/Post-intervention (95% CI)
Above a healthy weight (school)		
Eating vegetables	103	0.18 (−0.03 to 0.38)
Eating sweet and/or savoury snacks	81	−0.24 (−0.60 to 0.13)
Physical activity	120	0.46 (−1.10 to 2.01)
Sedentary behaviour	123	0.48 (−0.07 to 1.03)
Eating together as a family	105	0.04 (−0.14 to 0.22)
Eating in front of the TV	166	−0.07 (−0.38 to 0.24)
Above a healthy weight (community)		
Eating vegetables	28	0.32 (−0.29 to 0.93)
Eating sweet and/or savoury snacks	18	0.50 (−0.71 to 1.71)
Physical activity *	42	−3.26 (−6.41 to −0.11)
Sedentary behaviour	36	0.08 (−0.91 to 1.07)
Eating together as a family	29	−0.45 (−0.96 to 0.07)
Eating in front of the TV *	26	−0.85 (−1.58 to −0.14)
Healthy Weight		
Eating Vegetables	174	0.07 (−0.07 to 0.21)
Eating sweet and/or savoury snacks	153	0.17 (−0.06 to 0.40)
Physical activity *	180	−2.16 (−3.07 to −1.26)
Sedentary behaviour	170	−0.12 (−0.54 to 0.30)
Eating together as a family	187	0.05 (−0.12 to 0.22)
Eating in front of the TV	166	0.06 (−0.13 to 0.25)

* indicates *p*-value of <0.05.

## Data Availability

The data presented in this study are available upon request from the corresponding author.

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
