# Peer review of "A Natural Experiment Comparing the Effectiveness of the “Healthy Eagles” Child Weight Management Intervention in School Versus Community Settings"

_nutrients, 2021, doi:10.3390/nu13113912_

Round 1
Reviewer 1 Report
The encouraging result from this study is that, in the school setting, mean was down in all subgroups above healthy weight. This may point to an ongoing effort in schools to continue the program (I understand that this is not trivial to do) in some form having a long-term impact on public health.
Participants were recruited through the National Child Measurement Programme 150 (NCMP), self-referred or referred by GPs, health care professionals or their school. All except the last were community participants? If so, they already had a designation before the onset of the study? Therefore, less chance of regression to the mean? Since intent to treat wasn’t done, perhaps a sensitivity analysis could be done comparing baseline data between finishers, dropouts and incomplete. Similarly, of the ones who didn’t have 70+% of the intervention could be looked at, and a dose response analysis could be done.
healthy weight (z-score -1.34 to 1.34), overweight (z-score 1.34 to 2.05), obesity (z-score 2.05 to 206 2.65), severe obesity (z-score >2.65) Please fix overlapping ranges
The analysis for primary outcome was a paired t-test, result: not significant. Multiple linear regression was used to compare outcomes between the school-based 208 and community-based intervention with the community-based intervention as the reference. This wasn’t an ANCOVA? Multiple dummy variables? The paired t-test analyses didn’t account for the other covariates? Perhaps including covariates in the sub analyses would be a useful thing to do.
deprivation was sub-categorised into the two least deprived and the two most deprived centiles: So, only 4 percent of the population was used for this analysis? Or deciles? Please correct typos/clarify the deprivation analyses throughout the manuscript
Mean PA and eating in front of TV went down in 42 community respondents! Mean PA went down in 180 HW respondents (an apparently large effect!) Does this need a comment in the discussion?
Reviewer 2 Report
The reviewed paper provides an interesting example of an weight management intervention program conducted in two habitats - the school setting and the community setting. The purpose of this study is to assess the change in BMI z-score, and associated health behaviours, of children who attended the Healthy Eagles child weight management program and to compare these changes in the school versus the community setting. As well as to examine what, if any, outcome differences were observed in relation to gender, ethnicity, age or baseline weight status. While the paper is definitely very intriguing it is my opinion that it needs to be reorganized in order to be taken further in the journal editorial process.
Introduction
- This section is dominated by content regarding information about the Healthy Eagles program. It makes up ¾ of the entire introduction. In my opinion, the authors should have expanded the introduction to include examples of other interventions used in school and community-based settings, information on the differential effectiveness of these interventions based on scientific evidence. Also, since the authors used a natural experiment design they should justify why they designed the program this way with a citation of similar interventions undertaken by other researchers.
- The extensive section on the Heaty Eagles program should be condensed in the introduction. Excerpts can be included in the description of the study design.
- Lines 48-52 - I agree, there is evidence that programs targeted to the general public can be more effective for overweight and obese children than when targeted to specific groups. For example, a lifestyle intervention delivered to 15-year-old girls in Poland showed positive effects in a group of overweight and obese girls, although the programme was targeted at the general population (Dzielska, A., Mazur, J., Nałęcz, H., Oblacińska, A., & Fijałkowska, A. (2020). Importance of Self-Efficacy in Eating Behavior and Physical Activity Change of Overweight and Non-Overweight Adolescent Girls Participating in Healthy Me: A Lifestyle Intervention with Mobile Technology. Nutrients, 12(7), 2128. https://doi.org/10.3390/nu12072128).
- The purpose of the study is stated precisely. I have no comments.
Material and methods
This section should be expanded:
- Please describe the principles of conducting a natural experiment and justify why this type of natural experiment, conducted among children, is not subject to ethics committee review and, for example, consent from parents is not required. Please also provide references.
- Please present in detail, in a logical order, the assumptions of the Healthy Eagles program.
- The authors describe the theoretical basis - the program is based on scientific theories concerning forming and changing health behaviours, which I think is a very good point. They also list the topics of the classes, which is also a good aspect of the description.
- However, I miss a clear and precise listing of the methods used in the programme and which methods were used depending on the age of the participants (participants were of very different age, 4-16 years old). It is also not clear to me whether the program was identical in the school environment in the community? If not then please write what made these programs different?
Example: In one place the authors write that the program did not involve parents in any way, "There was no parental involvement in the intervention (lines 104-105, and in the next that "The community program was developed based on a "rolling, drop-in" design, meaning that families could start immediately after referral instead of waiting for an appropriate program to begin. This offered flexibility for families unable to attend every week and had the added benefit of allowing participants just starting their journey to learn from those further along in the program creating a peer-to-peer mentoring model." However, which indicates that parents and even entire families were involved in some way. Please be clear about who was a participant in the program. If it was only children - how they managed some tasks without parents' involvement, e.g. in this situation: Participants were also given a guided journal and asked to fill this in weekly to support their learning at home (lines 103-104).
Example 2: It is not clear how long the program lasted: 10 weeks (e.g. lines 57, 354) or 2-3 months (line 165).
- In my opinion, the text in lines 232-235 and Fig.1 "Participants flow chart" should be moved to the participants section.
- The schemes differed in their inclusion criteria, so it is difficult to say whether the better effect of the school-based programme was due to the fact that the programme took place in this environment or that its activities were aimed at the whole population rather than a selected group of children (with excess body weight), as in the case of the community-based programme.
- Lines 181-187 - described how information about different behaviours was obtained .I reiterate my question about how the youngest participants, children aged 4,5 or 6, coped with answering such a question?
- What were the questions asked of the participants? Were they asked about the frequency of consumption of selected foods? What type of physical activity was asked about? Moderate-to-vigorous or vigorous?
Results
Line 275 - should be Table 5 instead of Table 4
Discussion
The authors do not in any way explain what the unfavourable changes in physical activity or favourable changes in TV eating might have been due to or why there were no changes in other areas. Please feel free to complete the dissertation.
